

**Significant secondary formation of nitrogenous organic aerosols in an urban atmosphere revealed by**
**bihourly measurements of bulk organic nitrogen and comprehensive molecular markers**
Xu Yu[1], Min Zhou[2], Shuhui Zhu[2], Liping Qiao[2], Jinjian Li[3], Yingge Ma[2], Zijing Zhang[1], Kezheng Liao[3],
Hongli Wang[2], Jian Zhen Yu[1,3,4,*]
[1]Division of Environment and Sustainability, Hong Kong University of Science & Technology, Clear Water
Bay, Kowloon, Hong Kong, China
[2]Key Laboratory of Formation and Prevention of Urban Air Pollution Complex, Ministry of Ecology and
Environment, Shanghai Academy of Environmental Sciences, Shanghai, China
[3]Department of Chemistry, Hong Kong University of Science & Technology, Clear Water Bay, Kowloon,
Hong Kong, China
[4]Fok Ying Tung Graduate Research Institute, Hong Kong University of Science & Technology, Nanshan,
Guangzhou, China
Corresponding author: jian.yu@ust.hk
**Key points:**
1. Various primary and secondary sources of aerosol organic nitrogen (ON) have been quantitatively resolved
using PMF model based on bihourly measurement data of bulk ON and comprehensive source markers

2. Observational evidence of formation of reduced ON species through ammonia chemistry was found

3. Joint source analyses of ON and organic carbon (OC) facilitated investigating the potentially significant
formation pathways of ON aerosols



**Abstract**
Nitrogenous organic aerosol (OA) has a significant impact on solar radiation, human health, and ecosystems.
However, our knowledge of the total budget of aerosol organic nitrogen (ON) and its major sources,
particularly the secondary formation processes, remains largely qualitative. In this study, we conducted
bihourly measurements of aerosol ON and a comprehensive array of organic and inorganic source markers
in urban Shanghai during the fall-winter period of 2021. ON accounted for 6-58% of the total aerosol N,
averaging 20%. Positive factorization matrix source apportionment revealed that both primary emissions
(52%) and secondary formations (48%) made substantial contributions to the ON mass. Dominant primary
ON sources included coal combustion and vehicle emissions, accounting for 21% each. Five significant
secondary formation processes involving ON formation were identified, namely nitrate formation (14%),
photochemical formation (10%), nitroaromatics formation (7%), dicarboxylic acids (DCA) formation (8%),
and oxygenated cooking OA (7%). DCA formation-related ON likely represented reduced N-containing
organic species such as imidazoles and amides. Nitrate formation processes produced OA with a very low
organic carbon-to-ON ratio, suggesting a heterogeneous/aqueous formation of organic nitrates. Our field
work provides first quantitative source analysis and new insights into the secondary formation processes of
ON aerosols in an urban atmosphere.

**Key words**
Nitrogenous organic aerosol; Secondary formation processes; Bulk organic nitrogen; Source apportionment



**1 Introduction**

Nitrogen (N)-containing organic compounds are significant constituents of ambient organic aerosols (Yu et al., 2024), and their environmental effects have been observed in various aspects. For example, nitroaromatic compounds (Laskin et al., 2015; Xie et al., 2017) and imidazole-like species (Bones et al., 2010; Li et al., 2019) are typical brown carbon molecules that absorb solar radiation, leading to a warming effect. Amines are more efficient than ammonia ($NH_3$) in reacting with sulfuric acid to form new particles (Qiu et al., 2011), even in urban regions with high aerosol loading (Yao et al., 2018). Nitro-polycyclic aromatic hydrocarbons (nitro-PAHs) are known toxicants to the human body (Miller-Schulze et al., 2010; Bandowe et al., 2017). Additionally, the atmospheric deposition of organic nitrogen (ON) species serves as a significant source of N nutrient for marine and remote continental regions (Kielland et al., 2006; Andersen et al., 2017; Li et al., 2023). Therefore, detailed investigations are warranted to understand the budgets and sources of ON aerosols considering their multiple important environmental effects.

Several compound categories of ON have been identified, including urea (Mace et al., 2003; Violaki and Mihalopoulos, 2011), amino acids (Zhang et al., 2002; Ren et al., 2018), amines (Ho et al., 2016; Liu et al., 2018a), N-heterocyclics (Samy and Hays, 2013; Rizwan Khan et al., 2017), nitroaromatics (Chow et al., 2016; Xie et al., 2017), nitro-PAHs (Wei et al., 2012), and organic nitrates (Li et al., 2018; Huang et al., 2021b). While urea stands as a single compound, several to dozens of individual compounds have been quantified in each of the other categories. Despite considerable quantification uncertainty, aerosol mass spectrometry (AMS) has been widely used to estimate the total amount of organic nitrates (Farmer et al., 2010; Huang et al., 2021b; Xu et al., 2021). Overall, the quantifiable individual ON species or a specific ON category commonly constitute only a minor fraction of the total ON aerosol content (Jickells et al., 2013). The comprehensive quantification of every ON molecule to derive the total ON aerosol budget is impractical due to the lack of knowledge of molecular composition of the ON fraction and standards. Alternatively, bulk ON measurement, though lacking detailed compositional data, enables mass closure and aids in exploring major sources of ON aerosol. Traditional methods of aerosol ON quantification have relied on the difference method, where ON is calculated as the difference between total nitrogen (TN) and inorganic nitrogen (IN) (Cape et al., 2011). Limitations with the traditional analytical approach for aerosol ON determination have led to three deficiencies in the current status of aerosol ON data.

First, the assessment of aerosol total ON, including both water-soluble ON (WSON) and water-insoluble ON (WION), has been quite restricted, with most determinations focusing solely on water-soluble TN (WSTN), omitting WION measurements (Cape et al., 2011). This approach produces WSON through taking the difference between WSTN and IN. Some studies have employed elemental analyzers for TN determination, calculating ON as the difference between TN and IN (Duan et al., 2009; Miyazaki et al., 2011; Pavuluri et al., 2015; Matsumoto et al., 2019). However, the elemental analyzers' detection limit of nitrogen is insufficient for accurate measurements of trace-level aerosol nitrogen (Duan et al., 2009), limiting its widespread use in aerosol nitrogen analysis. Despite significant uncertainty, a few studies suggested that WION, deduced by subtracting WSTN from TN, could be more abundant than WSON in coastal or urban areas (Pavuluri et al., 2015; Matsumoto et al., 2019), highlighting the necessity of quantifying total ON to



determine the extent of ON aerosol presence. Second, the quantification of both WSON and WION using
the difference method introduces considerable uncertainty, especially when ON is a minor fraction of TN
(Yu et al., 2021). This approach has led to the reporting of physically implausible negative WSON
concentrations in past studies (Mace et al., 2003; Nakamura et al., 2006; Violaki and Mihalopoulos, 2010; Yu
et al., 2017). Third, absence of high-time resolution or online measurement methods for aerosol ON has
hampered the investigation of ON aerosol sources and formation processes in previous research.

We have developed an analyzer system that utilizes programmed thermal evolution of carbonaceous and

nitrogenous aerosols and chemiluminescence detection coupled with multivariate curve resolution data
treatment (Yu et al., 2021). This system enables simultaneous quantification of aerosol IN and ON with high
sensitivity and accuracy. Unlike conventional methods, our new approach avoids the occurrence of negative
ON concentrations, which are often encountered in difference methods. Furthermore, the method allows for
both offline and online measurements of aerosol ON. During the summer of 2021, we conducted a two-
month period of online observations of aerosol IN and ON in urban Shanghai (Yu et al., 2023). Our findings
revealed significant diurnal variations in ON concentrations, with vehicle emissions and secondary
formation processes identified as major drivers of episodic ON enhancements. However, due to the lack of
comprehensive organic source markers, we were unable to fully attribute the contributions of certain
potentially important sources and/or formation processes to the ON budget.

In this study, we extended our investigation through online measurements of aerosol ON in urban

Shanghai during the fall-winter period of 2021. Concurrently, we conducted comprehensive measurements
of aerosol major components and source markers on an hourly/bihourly scale. Specifically, we measured a
comprehensive array of organic tracers representing distinct primary emission sources and secondary
formation processes. These measurements enabled us to quantitatively apportion total ON to different
primary and secondary sources using positive matrix factorization (PMF) receptor modeling. Our focus lies
in examining the secondary formation sources of ON, as our knowledge regarding the formation mechanisms
of ON aerosols remains limited. By combining the high-time resolution measurements of ON and
comprehensive organic markers, we demonstrate the successful quantitative source analysis of ON aerosols
in an urban atmosphere, revealing significant contributions of secondary formation pathways to ON.

**2 Methodology**
**2.1 Sampling site and period**
The field measurement was conducted in Shanghai, a megacity located in the Yangzi River Delta (YRD)
region of China and with a population of over 24 million. In recent years, the city has experienced frequent
episodes of $PM_{2.5}$ pollution, with nitrogenous components becoming increasingly prominent contributors to
$PM_{2.5}$ mass (Zhou et al., 2022). All measurements were carried out at a monitoring site (31.17°N, 121.43°E)
situated on the rooftop of an eight-story building, approximately 30 meters above the ground, at the Shanghai
Academy of Environmental Sciences (SAES). This site is surrounded by urban roads, commercial activities,
and residential dwellings, making it a representative urban location influenced by a diverse range of emission
sources (Wang et al., 2018; Zhou et al., 2022). The observations were conducted during the fall-winter period



from November 6 to December 31, 2021.
**2.2 Online measurement of aerosol ON**
Aerosol ON was measured bihourly using our newly developed analytical system, which enables sensitive
and simultaneous measurements of aerosol ON and IN. Detailed descriptions of the new method can be
found in our previous work (Yu et al., 2021; Yu et al., 2023). In brief, the analyzer system integrates two
commercial instruments: an online aerosol carbon (C) analyzer and a chemiluminescence $NO_x$ analyzer.
Carbonaceous and nitrogenous aerosols collected on quartz filters are thermally evolved under programmed
temperatures and then catalytically oxidized to $CO_2$ and nitrogen oxides ($NO_y$), respectively. The C signal is
monitored using the non-dispersive infrared (NDIR) method, while the N signal is recorded through
chemiluminescence detection after converting $NO_y$ to NO. The C signal assists in differentiating IN and ON
components since ON aerosols produce both C and N signals, while the IN fraction only gives an N signal.
The programmed thermal evolution facilitates the separation of aerosol IN and ON, as they exhibit distinct
thermal characteristics. The quantification of IN and ON is achieved through multivariate curve resolution
data treatment of C and N thermal fractions (Yu et al., 2021).
The time resolution for ON measurement is 2 hours, with each sampling lasting one hour, followed by an
analysis step taking around 50 minutes. Sampling commenced at even hours (e.g., 02:00, 04:00). In total,
598 pairs of available ON and IN data points were collected. 4-methyl-imidazole was used as the standard
for calibrating C and N measurements, with calibration performed twice a month. The detection limit for
aerosol N is 0.013 μgN, corresponding to an air concentration of 0.027 μgN m$^{-3}$.
**2.3 Other online measurements**
The measurement methods for PM$_{2.5}$ mass and major aerosol components at the site have been described in
detail elsewhere (Qiao et al., 2014). PM$_{2.5}$ concentration was measured using a beta attenuation particulate
monitor (Thermo Fisher Scientific, FH 62 C14 series). Organic and elemental carbon (OC and EC) were
monitored using a semicontinuous OC/EC analyzer (model RT-4, Sunset Laboratory, Tigard, OR, USA). The
major water-soluble ionic species ($NO_3^-$, $Cl^-$, $SO_4^{2-}$, $Na^+$, $NH_4^+$, $K^+$, $Mg^{2+}$, and $Ca^{2+}$) in PM$_{2.5}$ were measured
using the Monitor for AeRosols and GAses (MARGA) (ADI, 2080; Applikon Analytical B.V.). Elements in
PM$_{2.5}$ (e.g., Al, K, Ca, Mn, Fe, Ni, Cu, Zn, As, Se, Cd, Pb) were monitored using an online x-ray fluorescence
(XRF) spectrometer (Xact® 625, Cooper Environmental Services, Tigard, OR, USA). All the instruments
were equipped with individual sampling inlets with a cyclone to achieve a 2.5 μm cut size. The sampling
lines, made of stainless steel, were approximately 2-2.5 m in length.
Quantification of a suite of speciated organic markers was conducted using a Thermal desorption Aerosol
Ggas chromatography–mass spectrometer (TAG, Aerodyne Research Inc.). The measurement principle and
operational procedure of the TAG system have been described in detail in previous studies (Williams et al.,
2006; He et al., 2020; Zhu et al., 2021). In brief, the TAG system operated at a time resolution of 2 hours.
During the first hour, ambient air was drawn through a PM$_{2.5}$ cyclone at a flow rate of 10 L min$^{-1}$, passing
through a carbon denuder to remove the gas phase, and particles were then collected onto a thermal
desorption cell (CTD). In the second hour, the collected particles underwent thermal desorption and gas
chromatography–mass spectrometry (GC-MS) analysis. In each analysis, 5 μL of an internal standard



mixture was added to the CTD that was loaded with particles collected in the preceding hour. During the
thermal desorption step, the polar organic compounds in the $PM_{2.5}$ were derivatized to their trimethylsilyl
derivatives under a helium stream saturated with the derivatization agent N-methyl-N-(trimethylsilyl)
trifluoroacetamide (MSTFA). Subsequently, the organic compounds were reconcentrated onto a focusing
trap cooled by a fan. Following this, the CTD was purged with pure helium to remove excess MSTFA, and
the focusing trap was heated to 330°C to transfer the organic compounds into the valveless injection system,
which utilizes a restrictive capillary tube to connect to the GC inlet. The GC-MS analysis was then initiated.
A total of around 100 polar and nonpolar organic compounds could be identified and quantified with
authentic standards (Zhu et al., 2023). The individual TAG-measured source tracers used for PMF receptor
modeling are listed in Table S1.

Gaseous pollutants, including sulfur dioxide ($SO_2$), ozone ($O_3$), nitrogen dioxide ($NO_2$), nitric oxide (NO),

and carbon monoxide (CO), along with meteorological parameters such as temperature, relative humidity,
atmospheric pressure, visibility, wind speed, and wind direction, were also recorded. The trace gas and
meteorological data were averaged to an hourly resolution to match the time resolution of other analyses.

**3 Results and discussion**
**3.1 Abundance and variations of ON aerosol.**
The concentration of aerosol ON ranged from 0.15 to 2.35 µgN m$^{-3}$, with the average being 0.80±0.45 µgN
m$^{-3}$ during the fall-winter period of observation (Figure 1). IN concentration was on average 3.34±2.16 µgN
m$^{-3}$ and displayed a large variation from 0.34 to 21.05 µgN m$^{-3}$. ON accounted for 6-58% of the total N in
aerosols, with an average of approximately 20%. This percentage was significantly lower than ($p < 0.001$)
the value (25%) observed at the same location during the summer of 2021 (Yu et al., 2023). The difference
was attributed to the higher enhancement of IN (2.2 times) compared to ON (1.8 times) from summer to
winter. During the winter observation, ON exhibited significantly higher concentrations at nighttime (0.85
µgN m$^{-3}$) compared to daytime (0.75 µgN m$^{-3}$) ($p < 0.005$) (Figure 1). This result contradicted the findings
from summertime measurements, which showed higher ON concentrations during daytime (Yu et al., 2023).
From summer to winter, the daytime and nighttime ON concentrations increased by a factor of 1.56 and 2.13,
respectively, suggesting that nocturnal formation of ON aerosols might be more significant in winter. The
concentration of aerosol IN was significantly higher during nighttime in both summer and winter periods at
the urban site.

The average of OC/ON atomic ratio was 8.15, with an interquartile range of 6.92 to 9.34 during the fall-

winter observation (Figure 1). That is, in most cases there was one N atom relative to 7-9 C atoms in the
collected organic aerosols. The average of OC/ON atomic ratio was significantly lower in winter than in
summer (8.15 vs 11.98) ($p < 0.001$). Since the major primary sources of OC and ON did not change much
at the urban site over seasons, which is discussed below, the lower OC/ON ratio suggested that during
wintertime, when air pollution is more severe, more N element was incorporated into organic molecules to
form secondary N-containing OA. Therefore, detailed investigations into the sources, particularly the
secondary formation processes of nitrogenous OA, are warranted.



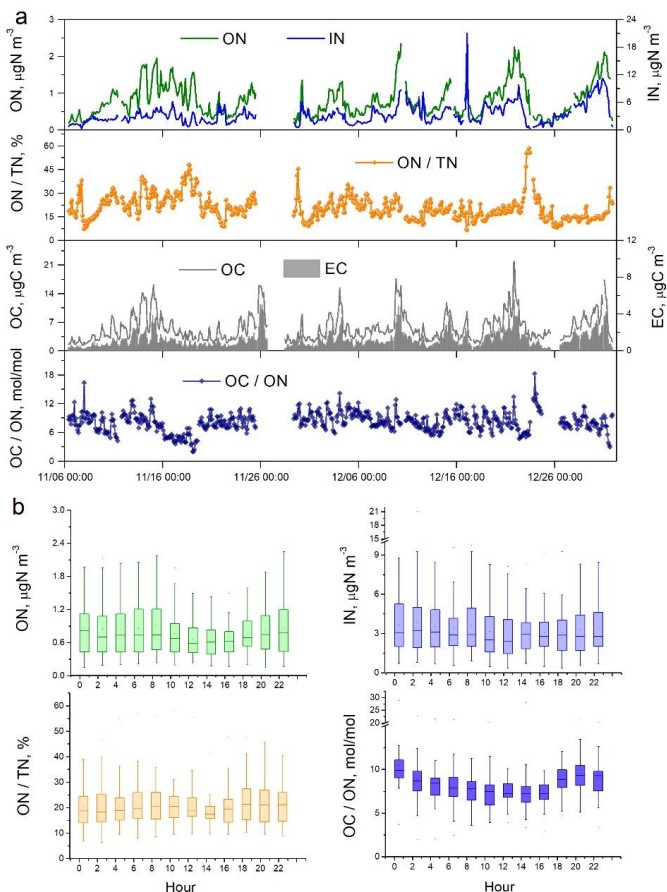

**Figure 1**. (a) Time series of aerosol N and C concentrations as well as ON/TN and OC/ON ratios during the fall-winter field observations in urban Shanghai from November 06 to December 31 of 2021.The OC/ON values are atomic ratios, calculated by measured OC/ON mass ratios divided by (12/14). (b) Diel variations of ON and IN concentrations as well as ON/TN and OC/ON ratios.

**3.2 Source apportionment for aerosol ON by PMF model with comprehensive source markers.**

In this study, bihourly measurements of aerosol total ON together with a comprehensive list of organic and inorganic source markers were available. This allows us to quantitatively resolve the various sources of ON using PMF receptor modeling. The diel variations of ON sources can be revealed from the model results. The descriptions of PMF run are provided in Text S1 in Supporting Information. Benefiting from the comprehensive array of molecular source tracers, the PMF analysis resolved a total of 18 factors, including 8 primary emission sources and 10 secondary formation sources. The factor profiles and contributions are displayed in Figure S2 and Figure S3, respectively. The 18 factors were identified by the characteristic tracers in individual source profiles (Qiao et al., 2014; Wang et al., 2015), which were: (1) manganese (Mn), iron (Fe), and zinc (Zn) for industrial emissions; (2) selenium (Se), and lead (Pb) for coal combustion; (3) levoglucosan, mannosan, and galactosan for biomass burning; (4) hopanes, $NO_x$, and EC for vehicle emissions; (5) nickel



(Ni) for ship emissions; (6) saturated and unsaturated fatty acids for cooking emissions; (7) azelaic acid, 9-
oxononanoic acid, and nonanoic acid for oxygenated cooking OA formation. This factor shows the profile
of oxidation products from cooking emissions. It could be found that both cooking emissions and oxygenated
cooking OA showed significantly enhanced contributions during dinner time (Figure S3b); (8) sodium ($Na^+$)
and chloride ($Cl^-$) for sea salt emissions; (9) silicon (Si) and calcium (Ca) for soil dust; (10) nitrocatechols
for nitrocatechol formation processes. Nitrocatechols are a combination of 4-nitrocatechol, 3-methyl-5-
nitrocatechol, and 4-methyl-5-nitrocatechol. These species were significantly correlated with each other,
with R-squared values ranging from 0.4 to 0.8; (11) 4-nitrophenol for nitrophenol formation processes. Note
that nitroaromatic compounds are notable constituents of nitrogen-containing OA, which leads us to include
nitrocatechols and nitrophenol in the PMF analysis to resolve ON fraction linking with nitroaromatic
components; (12) nitrate ($NO_3^-$) for nitrate formation processes; (13) sulfate ($SO_4^{2-}$) for sulfate formation
processes; (14) $O_3$ for photochemical formation processes; (15) phthalic acid for phthalic acid formation
processes; (16) dicarboxylic acids (DCAs) for DCA formation processes; (17) isoprene and α-pinene SOA
tracers for isoprene & α-pinene SOA formation processes; and (18) β-caryophyllenic acid for β-
caryophyllene SOA formation processes. The 18-factor solution exhibited excellent agreement with the
observed ON and OC masses (Figure S4). 12 out of the 18 factors have contributed to the ON mass, while
no ON was distributed in the remaining 6 factors including biomass burning, ship emission, sea salt emission,
sulfate formation, phthalic acid formation, and isoprene & α-pinene SOA formation processes.
Overall, 52% (0.42 μgN m$^{-3}$) of the ON mass was derived from primary emissions (Figure 2). Coal
combustion and vehicle emissions were the two dominant primary sources of ON, each contributing
approximately 20% to the aerosol ON. This finding was consistent with the summertime measurements at
the same site (Yu et al., 2023). The contribution of coal combustion to ON was higher during nighttime,
while the contribution of vehicle emissions was enhanced during rush hours (Figure 2c). Industrial and dust
emissions had relatively lower contributions to ON but showed increased contributions during daytime.
Unlike many studies reporting biomass burning as a significant source of aerosol ON (Mace et al., 2003;
Chen and Chen, 2010; Yu et al., 2017), we found that biomass burning made a negligible contribution to the
ON pool in urban Shanghai during our observation period. Note that the observed concentrations of biomass
burning markers, including levoglucosan, mannosan, and galactosan, displayed small day-to-day variations
(Figure S6). Their upward and downward trends were mainly driven by diel changes, which can be
influenced by differences in degradation rates and boundary layer heights between day and night. The
contribution of the PMF-resolved biomass burning factor remained relatively constant over the observation
period, with only one nighttime peak which might be due to an uncommon nighttime biomass burning event
occurred not far from the sampling site (Figure S3). These results indicate absence of influence of notable
biomass burning plumes during the entire observation. Consequently, a significant contribution of biomass
burning to aerosol ON was not observed. Biomass burning was also a minor contributor (4%) to organic
carbon (OC) (Figure S5), similar to previous studies suggesting that biomass burning only contributed 3-4%
to OC in urban Shanghai (Li et al., 2020; Huang et al., 2021a). However, levoglucosan showed a good
correlation with nitrocatechols (Figure S7), and a significant fraction of ON was associated with



nitrocatechol formation processes (Figure 2). This likely indicated that the aging of biomass burning aerosols present in the regional background air produced a number of nitrogen-containing organics, such as nitroaromatic compounds, which contributed significantly to the aerosol ON pool at the urban site. The contribution of primary cooking emissions to ON was generally minor (2%), but it increased to 5±4% (ranging from 0.8% to 17%) during dinner time (Figure 2).

Secondary formation processes contributed 48% (0.38 μgN m$^{-3}$) to ON in the urban atmosphere. Among these processes, a factor related to oxygenated cooking OA was identified through the PMF analysis. This source was characterized by the significant presence of azelaic acid, nonanoic acid, and 9-oxononanoic acid, which are oxidation products of oleic acid or other unsaturated fatty acids with a -C=C at C9 position that are emitted directly from cooking activities (Huang et al., 2021a; Wang et al., 2021). The presence of ON in the oxygenated cooking OA suggests the nitration of primary emitted cooking molecules or oxidation of N-containing molecules of primary cooking emissions (Sugimura et al., 2004; Zhao et al., 2011). Notably, this factor was found to contribute more to aerosol ON than primary cooking emissions. The percent contribution to total ON was 7% during the entire fall-winter observation period and increased to 13±9% during dinner time. When considering both primary cooking emissions and the oxygenated cooking OA, we found a significant contribution (9%) of cooking activities to ON aerosols in urban environments. This contribution was particularly pronounced during dinner time, reaching 17±10%. A substantial fraction (14%) of ON was derived from nitrate formation processes, which exhibited minimal diel variation. Photochemical formation processes represented 10% of the ON source, with a significant increase observed from noon to the afternoon. The formation processes of dicarboxylic acids (DCA) and nitrocatechols contributed 8% and 7%, respectively, to the ON budget.

ON associated with the factors related to nitrocatechol and nitrophenol formation likely represent the amount of N bound within nitroaromatic compounds, averaging approximately 60 ngN m$^{-3}$ (Figure 2). This study quantified four nitroaromatic compounds in aerosols, namely 4-nitrophenol, 4-nitrocatechol, 3-methyl-5-nitrocatechol, and 4-methyl-5-nitrocatechol, using the TAG system. The combined N content in the four nitroaromatic compounds averaged at 1.14 ngN m$^{-3}$, accounting for approximately 2% of the estimated total nitroaromatic-N. In most cases (>90%), the speciated nitroaromatic-N represented less than 10% of total nitroaromatic-N (Figure S8a). Nitroaromatic compounds are known for their significant contributions to aerosol light absorption, and some are recognized as toxicants (Laskin et al., 2015; Zhang et al., 2023). Our findings indicated the substantial presence of the un-speciated mass within this group of N-containing compounds, thus the importance to identify and quantify the unknown nitroaromatics.

We estimate the amount of oxidized ON by summing up the ON distributed in the factors related to atmospheric oxidation processes, including the photochemical formation, nitrate formation, and nitroaromatic formation factors. In this way, oxidized ON, mainly organic nitrates and nitroaromatics, had a concentration range of 0.02-0.85 μgN m$^{-3}$ with an average of 0.25 μgN m$^{-3}$. It accounted for 4-68% (25% on average) of total oxidized N (oxidized ON plus nitrate-N) in the aerosols (Figure S8b). The formation of oxidized ON and inorganic nitrate involves the common precursors of nitrogen oxides (NO$_x$). Our results suggested the yield of conversion of NO$_x$-N to aerosol ON and nitrate-N has an approximate ratio of 1:3.



This estimation, although rough, is valuable for evaluating the fate of NO$_x$ in urban areas. Our analyses
highlight the significance of measuring bulk ON to reveal the abundance of different groups of organic N-
containing aerosols. With comprehensive source markers available, the PMF analysis allows us to apportion
the total aerosol ON mass to different factors (sources), and each fraction of ON may indicate the total
quantity of N from the corresponding sources or formation processes.

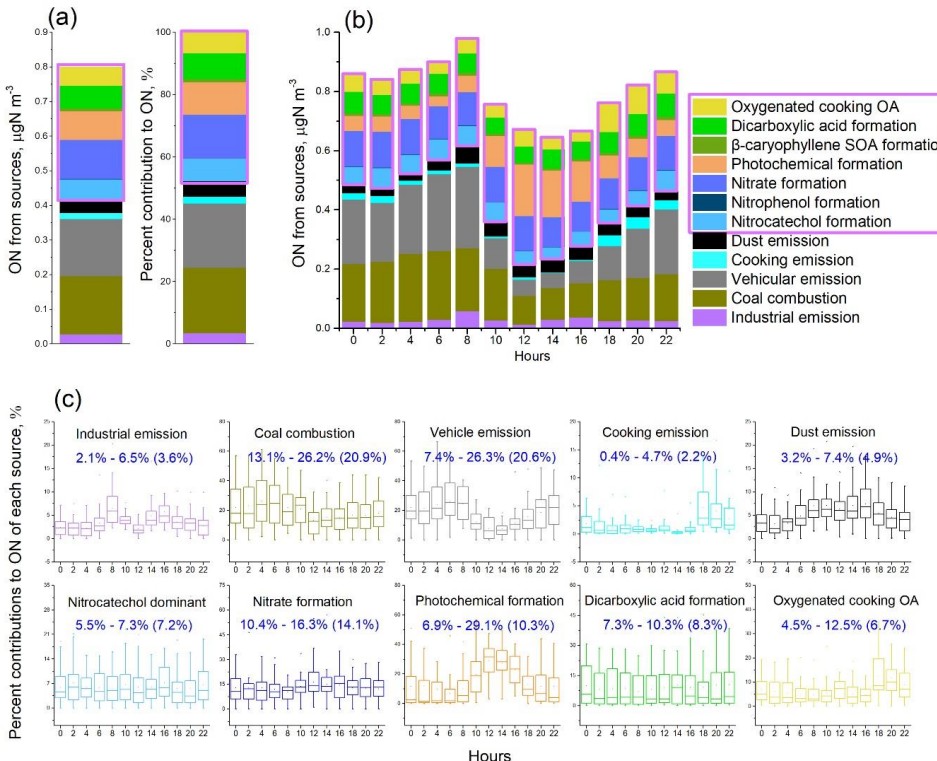

**Figure 2.** PMF source apportionment results for aerosol ON. (a) Overall mass and percent contributions of resolved sources to aerosol ON during the fall-winter observation period. (b) Diel variations of source compositions of ON. (c) Diel variation patterns of each ON source. The numerical ranges show the lowest to highest averages of percent contributions of sources at sampling hours (e.g., 2:00, 4:00). The number in parentheses are overall percent contributions of sources to ON pool during the observation (i.e., results in Figure (a)). Contributions of nitrophenol formation and β-caryophyllene SOA formation to ON were very minor and not shown for their diel variations in (c). Secondary sources of ON are highlighted with a purple box. The sources of OC can be found in Figure S5.

### 3.3 Evidence of formation of reduced ON species

It is noteworthy that a significant fraction of ON (8%) was associated with the formation processes of DCAs
(Figure 2). DCAs in ambient aerosols are primarily derived from the oxidation of anthropogenic and biogenic
volatile organic compounds. In this study, the concentration of DCAs increased with rising relative humidity
(RH) (Figure S9), suggesting a higher RH could enhance the aqueous formation of DCAs and/or the gas-to-
particle partitioning of DCAs. The part of ON related to DCA formation processes (here termed as DCA_ON)
may represent the reduced-ON species formed through the heterogenous/aqueous phase reactions between




DCAs and NH$_3$/NH$_4^+$, as discussed below. Previous lab studies and field measurements have suggested that
the amount of particulate NH$_4^+$, as measured by AMS, exceeded the quantity required to balance anions
including nitrate (NO$_3^-$), sulfate (SO$_4^{2-}$), and chloride (Cl$^-$). The excess NH$_4^+$ was believed to bind with
organic acids such as DCAs to form organic ammonium salts (Schlag et al., 2017; Hao et al., 2020). We note
that the measurement of NH$_4^+$ by AMS relies on the quantification of NH$_x^+$ fragments, which could also
originate from the fragmentation of other reduced ON species, such as amines and amides, in addition to
NH$_4^+$ and organic ammonium salts. Consequently, the specific molecules into which the excess NH$_4^+$-N is
incorporated remain unclear due to the lack of molecular information on ON-containing compounds.

We examined the relationship between DCA_ON and NH$_4^+$ concentrations during periods of continuous

increment in DCA_ON lasting 4 hours or longer. We identified 17 such cases during the field observation
period. Interestingly, DCA_ON showed a strong correlation with NH$_4^+$ in all these cases (Figure 3a). This
result lends support for the hypothesis that DCA_ON may represent reduced ON species formed through the
reactions between DCAs and NH$_3$/NH$_4^+$. The slopes of linear regression between DCA_ON and NH$_4^+$ varied
significantly among the cases and could be roughly divided into higher-slope cases and lower-slope cases,
with 7 and 10 in these two categories, respectively. (Figure 3a). A higher-slope value indicated a more rapid
formation of DCA_ON at a given concentration level of NH$_4^+$. The higher-slope cases were distinctly
associated with higher O$_3$ concentration (Figure 3b), suggesting an elevated oxidation capacity and hence
enhanced formation of secondary products such as DCAs, which serve as the precursors of DCA_ON.
Additionally, the higher-slope cases were associated with lower pH values (higher acidity) as calculated
using the thermodynamic equilibrium model ISORROPIA II with MARGA data (Figure 3b). The lower pH
facilitated the gas-to-particle partitioning of NH$_3$ and subsequent reactions involving DCAs and NH$_3$. Note
that the aqueous formation of imines such as imidazoles through the reactions between carbonyls and
NH$_3$/NH$_4^+$ has been established in laboratory studies (Galloway et al., 2009; Noziere et al., 2009) and
confirmed by field observations (Zhang et al., 2020; Lian et al., 2021; Liu et al., 2023). Considering the close
relationship between carbonyls and DCAs, the possibility that imines contributed to DCA_ON could not be
excluded. Overall, these results provided observational evidence of potentially significant formation of
reduced ON species through NH$_3$ chemistry in the real atmosphere.

The concentration of DCA_ON had an interquartile range of 8-111 ng N m$^{-3}$, with an average ± standard

deviation (SD) of 66 ± 81 ng N m$^{-3}$ over the observation period. Assuming an average molecular formula of
C$_5$H$_7$N$_{1.5}$O$_{1.5}$ for the reduced-ON species, considering these compounds may contain 3-7 carbon, 1-2
nitrogen, and 1-2 oxygen atoms, the concentration of these ON compounds would be 43-592 ng m$^{-3}$ with the
average ± SD being 352 ± 432 ng m$^{-3}$. This result provides a rough estimation of the total reduced ON
compounds that are formed through NH$_3$/NH$_4^+$ reactions in urban Shanghai. In a recent study conducted in
rural Shanghai (Liu et al., 2023), eight imidazoles were detected, and the total concentration of these species
ranged from 1.3-15.8 ng m$^{-3}$ (average: 5.5 ± 3.4 ng m$^{-3}$). Furthermore, significant increases in imidazole
concentrations were observed during humid haze periods, suggesting an aqueous phase formation pathway
of these species (Liu et al., 2023). The summed concentration level of the eight imidazoles was lower by 1-
2 orders of magnitude compared to our estimated bulk concentration, indicating the prevalence of





unidentified reduced ON species. Our analyses suggested that ON aerosols originating from $NH_3$ chemistry
could be a significant source of nitrogenous SOA. Further investigations are needed to determine their major
chemical compositions and formation mechanisms.

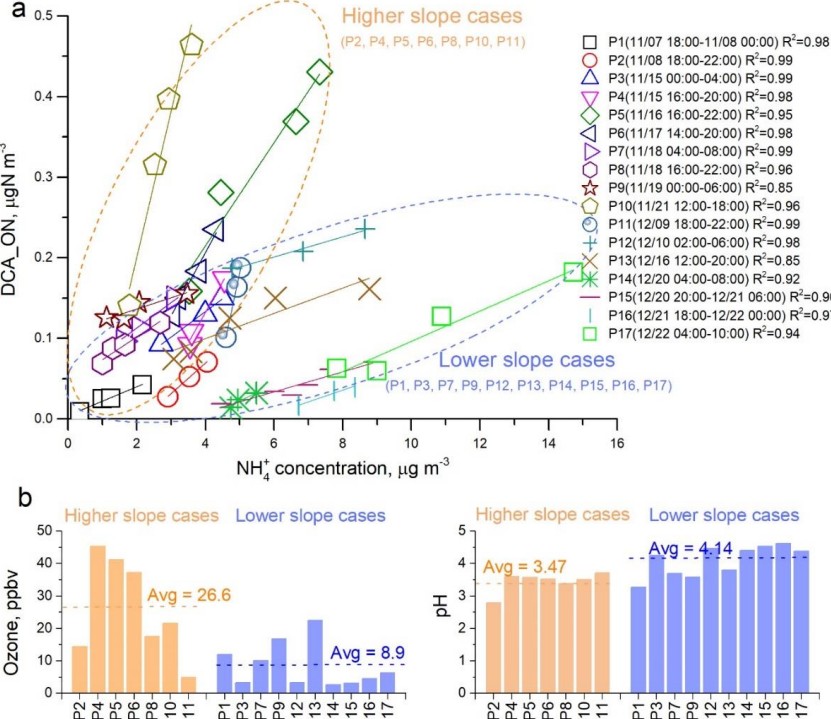


**Figure 3**. The relationship between ON associated with dicarboxylic acids (DCA) formation (DCA_ON)
and ammonium ($NH_4^+$) in the cases that DCA_ON showed continuous increments, and the potential
influence factors. (a) Correlations between DCA_ON and $NH_4^+$ concentrations in each of 17 cases where
when DCA_ON exhibited a continuous increment. The cases can be separated into two groups based on their
linear regression slopes, those with higher slopes, enclosed in yellow dashed circles, and those with lower
slopes, enclosed in blue dashed circles. (b) Comparisons of levels of ozone ($O_3$) and pH in higher slope cases
and lower slope cases.


**3.4 Factors driving the increment of secondary ON aerosol.**

The PMF source apportionment analysis has provided the total quantity of the secondary ON (SON) and its
apportionment to individual formation pathways. This greatly facilitates investigating the largely under-
evaluated formation processes of N-containing OA. We next focus on the cases where SON showed
continuous increment for a period of 4 hours or longer and discuss the driving factors behind these
increments. Forty-two cases were identified throughout the entire observation period. To isolate the local
formation of SON from the influence of transported air masses, we extracted cases with wind speeds lower
than 3 m/s as cases of local SON formation (Zhou et al., 2022). The following discussion will exclusively
focus on the local SON formation cases.



A total of 24 local SON formation cases were identified and classified into five types based on the
dominant formation pathway of SON, as revealed by the PMF analysis. Figure 4 illustrates the variations in
the sources of SON and secondary organic carbon (SOC) for the five types of local SON increment. The
ensuing discussion shows our paired measurements of bulk aerosol ON and OC, along with subsequent
source analyses, provide unique insights into the formation processes of N-containing OA.
Type 1 cases, totaling three, were characterized by DCA formation processes driving the increment of SON.
In one example of Type 1 case shown in Figure 4, SON increased by 0.33 $\mu$gN m$^{-3}$ from 16:00-22:00 on
November 16. During this period, DCA_ON increased by 0.27 $\mu$gN m$^{-3}$, accounting for 82% of the variation
in SON. Nitrate formation processes also contributed to the SON increment during this period, but to a lesser
extent. DCA formation processes were also the dominant source of SOC increase in this case, contributing
to the production of 1.05 $\mu$g C m$^{-3}$ of SOC and 80% of the change in SOC during the period. Note that the
atomic ratio of the change in SOC ($\Delta$SOC) to the change in SON ($\Delta$SON) associated with DCA formation
processes was 4.5. This ratio indicates that there was approximately 1 nitrogen atom for every 4-5 carbon
atoms in the SOA associated with DCA formation processes. The low $\Delta$SOC/$\Delta$SON ratio suggested the
formation of N-containing organic aerosols with low molecular weight and/or multiple N atoms. Considering
that some SOA compounds may not contain N atoms, the SON compounds formed through DCA formation
processes may have a C/N ratio lower than 4. Possible candidate species include imines, such as methyl- and
ethyl-imidazoles, and C2-C5 amides. This result supports the aforementioned hypothesis that DCA_ON
likely constitutes of reduced ON species formed through reactions between acids/carbonyls and NH$_3$/NH$_4^+$.
All three Types 1 cases occurred during the transition from daytime to nighttime (e.g., from 16:00-22:00).
This suggests that acids/carbonyls accumulated during the daytime through photochemical processes and
then entered the aqueous phase during nighttime, where they reacted with NH$_3$/NH$_4^+$ to form reduced ON
species under higher air humidity.
Type 2 cases, totaling seven, featured photochemical formation as the dominant source driving SON
increment. They mostly occurred during the morning to noon periods, suggesting formation of SON through
photochemical processes. In the Type 2 case presented in Figure 4, $\Delta$SON was 0.18 $\mu$gN m$^{-3}$ and $\Delta$SOC was
0.97 $\mu$g C m$^{-3}$. Photochemical formation contributed 0.1 $\mu$gN m$^{-3}$ of SON and 0.61 $\mu$g C m$^{-3}$ of SOC, yielding
a $\Delta$SOC/$\Delta$SON atomic ratio of 7.1. This result suggested significant formation of N-containing organic
molecules with a C/N ratio of 6-8. Examples of such species include nitroaromatic compounds and organic
nitrates. The frequent occurrence of Type 2 cases suggested an efficient formation of oxidized ON species
in urban areas.
Type 3 cases, totaling six, exhibit SON increase driven by oxidation of cooking emissions. Their
occurrences coincided with lunch and dinner hours. Take the case during 16:00-20:00 of December 4 as an
example, 0.16 $\mu$gN m$^{-3}$ of $\Delta$SON and 1.15 $\mu$g C m$^{-3}$ of $\Delta$SOC were formed, resulting in a C/N ratio of 8.4
for the SOA associated with the cooking emission oxidation. Chamber simulations have shown that SOA
produced through oxidation of cooking fumes can be more abundant than primary OA species from cooking
emissions (Liu et al., 2018b). However, the chemical compositions and underlying formation mechanisms
of N-containing organic aerosols remain uncertain. This study, for the first time, reveals the potentially



significant contribution of cooking oxidation to N-containing SOA in the real urban atmosphere. Future
efforts are recommended to direct towards investigating representative N-containing molecules formed
through cooking oxidation and their formation pathways.
Four Type 4 cases were identified, each characterized by the dominant contribution of nitrocatechol
formation to ON increase and all occurred during the daytime. Take the case between 06:00 and 10:00 on
December 19 as an example, the sum concentration of 4-nitrocatechol, 3-methyl-5-nitrocatechol, and 4-
methyl-5-nitrocatechol (Table S1) rose from 9.52 to 24.64 ng m$^{-3}$. This increase was attributed to secondary
formation processes of nitroaromatic compounds, instead of primary emissions, as evidenced by the flat or
decreasing trends of elemental carbon (EC) and nitric oxide (NO) during that period (Figure S10). The
increment in SON and SOC associated with the nitrocatechol formation factor was 0.18 µgN m$^{-3}$ and 1.48
µgC m$^{-3}$, respectively over the four-hour period (Figure 4), resulting in a ΔSOC/ΔSON atomic ratio of 9.6.
This portion of SON may represent the N content in nitroaromatic compounds formed through processes
analogous to nitrocatechol formation.
Four Type 5 cases were identified, each characterized with nitrate formation processing as the driving
source for the increase in SON. This fraction of SON may indicate the formation of organic nitrates, which
share common precursor of $NO_x$ with nitrate. Organic nitrates have long been recognized as significant
components of secondary organic aerosols in ambient air (Rollins et al., 2012; Perring et al., 2013). Organic
nitrate formation encompasses two main pathways: hydroxyl radical (OH)-initiated oxidation of
hydrocarbons in the presence of $NO_x$ during the day and nitrate radical ($NO_3$)-initiated oxidation of alkenes
during the night. Both pathways involve the formation of organic nitrates in the gas phase, followed by
partitioning to the particulate phase (Perring et al., 2013). This study, through integrated analyses of SON
and SOC, provides evidence suggesting that organic nitrates might also form through heterogeneous or
aqueous reactions. As depicted in Figure 4, the nitrate formation process produced 0.17 µgN m$^{-3}$ of ΔSON
and 0.43 µg C m$^{-3}$ of ΔSOC from the night of December 20 to the following morning, yielding a
ΔSOC/ΔSON atomic ratio of only 2.9. Gas-phase formation of organic nitrates followed by gas-to-particle
partitioning would not result in such a low C/N ratio. Therefore, a significant number of organic nitrates
might be formed through heterogeneous or aqueous reactions between organic compounds and $HNO_3/NO_3$,
enhancing the ON content while not affecting the OC content which is already present in the particle phase.
In a previous study, it was suggested that organic nitrates can be produced through non-radical reactions of
hydrated glyoxal and nitric acid in the aqueous phase (Lim et al., 2016). Xu et al. (2020) found that aerosol
liquid water promotes the formation of water-soluble ON, likely in the form of organic nitrate species.
Previous studies have identified an 80% underestimation of monoterpene hydroxyl nitrate by the GEOS-
Chem model, which considers both OH oxidation and $NO_3$ oxidation mechanisms of monoterpene (Li et al.,
2018; Zhang et al., 2021), indicating an incomplete understanding of the formation mechanisms of organic
nitrates. Our observational results, combined with previous investigations, suggest the need for further
exploration of the formation mechanisms of particulate organic nitrates, such as heterogeneous/aqueous
phase reaction processes.



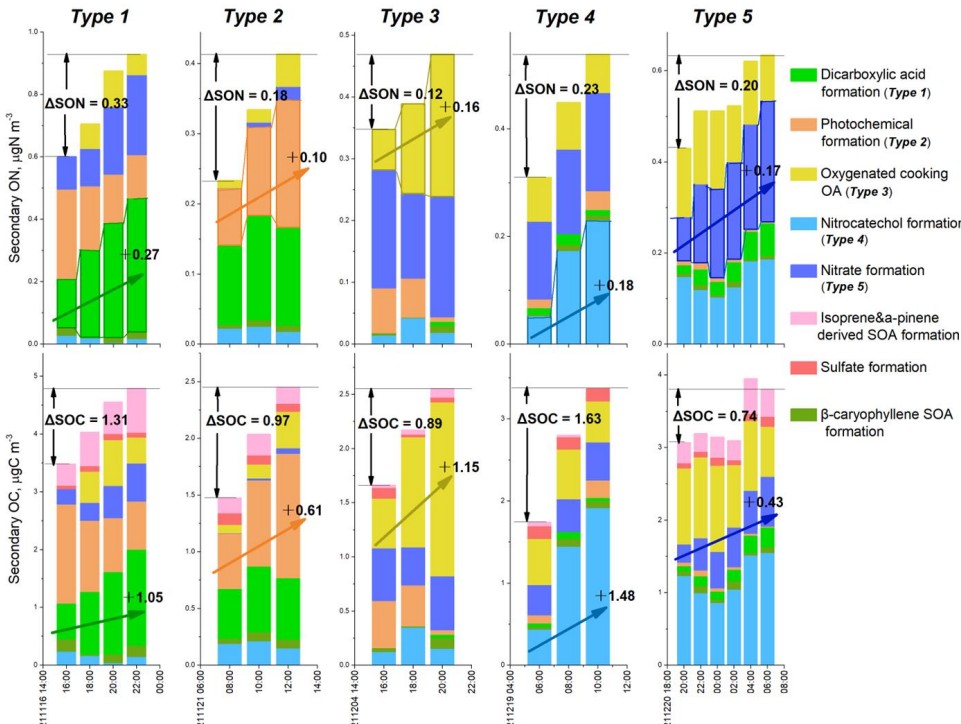

**Figure 4**. Five types of SON increment were driven by different formation processes. The increment amounts of SON (ΔSON) and SOC (ΔSOC) in each type are shown. The dominant formation processes of ON and OC in each type are highlighted with rising arrows.

## 4 Conclusion and implications

Presently, the knowledge of total aerosol ON budget is severely limited, with source analysis predominantly qualitative. Building on a methodological breakthrough that allows for the online measurement of bulk aerosol ON and concurrent measurements of a comprehensive array of molecular source tracers, we have identified both primary emissions and secondary formation processes as substantial contributors to $PM_{2.5}$ ON mass in urban Shanghai during the fall-winter period of 2021. While we acknowledge uncertainties of PMF modeling in apportioning sources to bulk ON, this approach facilitates identification of major sources/formation processes and provides quantitative insight into the relative importances.

The observed dominance of primary ON sources such as coal combustion and vehicle emissions, alongside the significant contributions from secondary formation processes like nitrate formation, photochemical processes, and DCA formation, indicates the multi-faceted nature of ON aerosol production in urban environments. The identification of specific secondary formation pathways, including nitroaromatics formation, DCA formation, and oxygenated cooking OA, sheds light on the diverse precursors and chemical processes involved in aerosol formation and evolution. Notably, we have provided valuable observational evidence on secondary ON aerosol formation through $NH_3$ and $NO_x$ chemistries, the joint evaluation of



which has been under-explored in the past.
The quantification of ON contributions from various sources and the elucidation of secondary formation
mechanisms provide a basis for targeted mitigation strategies aimed at reducing ON emissions and
improving air quality in urban areas. The insights gained from this study can inform policy decisions and
regulatory measures to curb primary emission sources and mitigate the impact of secondary formation
processes on ON aerosol levels.
Looking ahead, future research efforts should focus on refining our understanding of the detailed
mechanisms driving ON aerosol formation, including the chemical reactions involving major precursors and
secondary processes. Furthermore, continued monitoring and analysis of ON aerosol composition in
different environmental settings will be crucial for assessing the broader implications of ON aerosols on air
quality, climate, and public health. Bulk ON measurements enable mass closure and are advantageous for
constraining the major sources and formation processes of ON aerosols. This methodology complements the
molecular-level characterization of ON molecules, which provides chemical composition information but
falls short on capturing total ON. Future research efforts should emphasize identifying and quantifying ON
species that can indicate specific sources and formation processes.

**Financial support.** This work was supported by Science and Technology Commission of Shanghai
Municipality (20dz1204000), the Research Grants Council of Hong Kong (16213222 and 16304519).

**Author contributions.** XY and JZY conceived the research, designed the research plan and wrote the
manuscript; XY, MZ, SHZ, LPQ, JJL, YGM and HLW carried out the instrumental measurements; XY, ZJZ,
and KZL carried out data analysis.

**Competing interests.** The contact author has declared that none of the authors has any competing interests.

**Data availability.** Data used in this study is available upon request from the corresponding author.

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
