# Peer review of "Significant secondary formation of nitrogenous organic aerosols in an urban atmosphere revealed by bihourly measurements of bulk organic nitrogen and comprehensive molecular markers"

_EGUsphere, 2024_

## Referee Comment (RC2)

Comments

This manuscript presents an investigation into the sources and formation mechanisms of nitrogenous organic aerosols (ON) in an urban atmosphere, leveraging a new analytical technique for bulk ON quantification and high-resolution source apportionment. The study's strengths lie in its measurement approach, comprehensive source marker analysis, and identification of secondary ON formation pathways, including nitrate formation, photochemical oxidation, and DCA-associated reduced-N species. Several critical gaps in scientific rigor, data interpretation, and presentation weaken the manuscript's impact. Key concerns include incomplete source apportionment due to unclear presentation in methodology, insufficient mechanistic evidence for DCA_ON formation. Addressing these issues would significantly strengthen the manuscript's contribution to atmospheric chemistry and air quality research.

**1** The method used for total nitrogen estimation in this study is not commercially available. Additional details regarding the calibration procedure for total nitrogen quantification using this instrument should be provided. The IN and ON datasets should be cross-validated using independent measurements. For instance, it would be important to demonstrate whether water-soluble inorganic nitrate concentrations correlate strongly with the inorganic nitrogen values quantified in this study.

**2** The terms "Significant" and "significantly" are overused throughout the manuscript (lines 181, 339, 463, 353), yet no quantitative data or statistical evidence (e.g., percentage) are provided to substantiate these claims. For example, when stating that a result is "significant," the authors should clarify *how significant* it is (e.g., "a 30% increase with 95% confidence") rather than relying on qualitative assertions. Note that the term "significant" carries strong inferential weight in scientific writing; its use in the main text should be reserved for findings supported by rigorous statistical analysis.

**3** The assertion in line 194 that the primary sources of OC and ON "did not change across seasons" is questionable. For example, biomass burning and coal combustion are known to contribute seasonally variable OC/ON contributions. The authors should provide supplemental data or additional evidence (e.g., time-resolved source apportionment results) to support this claim in this study, particularly given evidence from prior studies demonstrating no seasonal shifts of ON and OC.

**4** The uncertainty associated with the 18 identified factors requires further verification. (a) The correlation matrix between factors should be presented to evaluate their interdependence.

(b) The observation that no organic nitrogen (ON) was attributed to biomass burning or isoprene/α-pinene SOA factors (lines 232-233) is unexpected and conflicts with extensive evidence. Biomass burning is a well-documented source of nitrogen-containing organic aerosols (OA) in both rural and urban environments.(Li et al., 2023) Similarly, biogenic VOC (BVOC) oxidation under high-NOx or NO3 radical-dominated conditions has been shown to produce ON(Xu et al., 2014). The authors should reconcile these discrepancies by either (1) re-examining their receptor modeling framework, (2) acknowledging limitations in source apportionment resolution.

 (c) The proposed formation pathway for ON from oxidized cooking emissions lacks mechanistic clarity and experimental support. The authors should either (1) provide chemical speciation data (e.g., HRMS)

linking cooking-derived organic compounds to ON precursors, (2) reference chamber studies demonstrating this pathway, or (3) propose a plausible reaction mechanism (e.g., radical-induced coupling of cooking-derived VOCs with NOx). Without such evidence, this conclusion remains speculative.

**5** The strong correlation between DAC_ON and $NH_4^+$ (line 324) requires further scrutiny. While demonstrating agreement between these variables is important, the authors must also rule out potential correlations with other PMF factors (e.g., Nitrate_ON) to confirm the specificity of this relationship.

**6** The statement in lines 393–396 that higher relative humidity (RH) promotes DCA_ON formation directly contradicts the evidence presented in Figure 3b, which shows that higher pH enhances DCA_ON formation. Higher RH usually leads to higher pH values.

**7** Methodology sections switch between past and present tenses (e.g., "We have developed... All measurements were carried out..."). Standardize to past tense for consistency.

References:
Li, Y., Fu, T.-M., Yu, J. Z., Yu, X., Chen, Q., Miao, R., Zhou, Y., Zhang, A., Ye, J., Yang, X., Tao, S., Liu, H., and Yao, W.: Dissecting the contributions of organic nitrogen aerosols to global atmospheric nitrogen deposition and implications for ecosystems, National Science Review, 10, 10.1093/nsr/nwad244, 2023.
Xu, L., Guo, H., Boyd, C. M., Klein, M., Bougiatioti, A., Cerully, K. M., Hite, J. R., Isaacman-VanWertz, G., Kreisberg, N. M., Knote, C., Olson, K., Koss, A., Goldstein, A. H., Hering, S. V., de Gouw, J., Baumann, K., Lee, S.-H., Nenes, A., Weber, R. J., and Ng, N. L.: Effects of anthropogenic emissions on aerosol formation from isoprene and monoterpenes in the southeastern United States, Proceedings of the National Academy of Sciences, 112, 37-42, 10.1073/pnas.1417609112, 2014.

---

## Author Response (AR1)

**Point-by-point response to review#1's comments**

Manuscript ID: egusphere-2024-4103

Title: "Significant secondary formation of nitrogenous organic aerosols in an urban atmosphere revealed by bihourly measurements of bulk organic nitrogen and comprehensive molecular markers"

Author(s): Xu Yu, Min Zhou, Shuhui Zhu, Liping Qiao, Jinjian Li, Yingge Ma, Zijing Zhang, Kezheng Liao, Hongli Wang, Jian Zhen Yu

We thank the reviewer for his/her constructive comments. Each of these comments has been addressed and detailed in our point-by-point response given below. The exact comment text from the reviewer is in black and italic style while our response text is in blue and normal format in this document. The revised texts are marked in blue in the highlighted manuscript. The line numbers mentioned below refer to those in the revised manuscript.

*The manuscript by Yu et al. presented high temporal resolution aerosol ON measurements and various source markers in urban Shanghai during the fall-winter period of 2021. The authors used the PMF model to identify up to 18 sources or formation processes of aerosol ON. Despite the potential limitations of the PMF predictions, the results obtained may provide valuable insights into the origins and formation processes of ON aerosols in Shanghai and serve as valuable references for future aerosol ON research. Overall, the manuscript is well-structured and well-written. I recommend that this paper could be published in Atmospheric Chemistry and Physics once the authors address the following minor comments.*

Response: We thank the reviewer for the positive comments.

*Major comments:*

*The author used the PMF model to identify up to 18 contributors of ON, which is very impressive. However, there seem to be some contradictions in the PMF results.*

*For example, 1) For factor nitrate formation, ammonium also has a high load. The liquid-phase reaction of ammonium is also one of the very important pathways for the formation of ON. Why is this factor only attributed to nitrate formation here;*

Response: The factors resolved in the PMF model were named based on the characteristically high loading of source indicators within each factor. Specifically, the nitrate formation factor was characterized by a high loading of nitrate. Ammonium was also abundant in this factor due to the neutralization reaction between ammonia and nitric acid.

In previous PMF analysis studies, researchers have referred to this ammonium nitrate-dominated factor using various names, such as "secondary nitrate" (Huang et al., 2021; Li et al., 2020), "secondary nitrate-rich" (Huang et al., 2018), "secondary nitrate formation processes" (Wang et al., 2019), and "nitrate-rich" (Yu et al., 2024). Despite the variations in naming, this factor consistently represents secondary formation processes characterized by significant nitrate production.

To enhance clarity and avoid ambiguity, we have revised the name of the "nitrate formation" factor to "nitrate formation processes" in the text and figures, emphasizing that this factor is primarily characterized by high nitrate loading while also containing other secondarily formed species. Similarly, we have updated "sulfate formation" to "sulfate formation processes" throughout the manuscript.

*2) For Figures 2 and S3b, the average contribution of traffic emissions to ON was higher from 0:00 to 8:00 (nighttime). Shouldn't the contribution be higher during the morning and evening rush hours?*

Response: While ON associated with vehicular emission during 0:00-8:00 was higher than other hours, we'd like to point out that the hour-to-hour variations within a 24-h cycle also indicated an elevated contribution of traffic emissions to ON during morning and evening rush hours. For example, shown in Figure 2, it is discernible that the contribution of vehicle emissions to ON peaked during 6:00-8:00 and 18:00-22:00, corresponding to morning and evening rush hours. The higher contribution of vehicle emissions at night may be attributed to the generally lower nighttime planetary boundary layer height, although the overall intensity of vehicle emissions at night might be lower compared to rush hours.

Overall, our results support the argument that vehicle emissions contributed more significantly to ON aerosols during morning and evening rush hours. We have explicitly stated in the manuscript that "the contribution of vehicle emissions was enhanced during rush hours." (Line 247-248)

*In addition, the average contribution of cooking and oxygenated cooking OA to ON were higher from 18:00 to 22:00. There should also be cooking at noon, and the photochemical reaction should be stronger, leading to more oxygenated cooking OA production;*

Response: As shown in Figure 2, there was a small but visible peak of contributions of cooking emissions and oxygenated cooking OA to ON around noontime (12:00), indicating enhanced cooking emissions and subsequent oxidation during lunchtime. The much lower contributions of cooking emissions and oxygenated cooking OA to ON at lunchtime compared to dinnertime might be attributed to 1) Lower cooking emissions at lunchtime compared to dinnertime; 2) More favorable dispersion conditions at noontime, which effectively dilute cooking emission plumes, leading to a lower contribution of cooking emissions and oxygenated cooking OA to the observed ON. A previous field study conducted in urban Shanghai also observed a small peak of cooking emission at noontime, with substantial increase during dinnertime (Huang et al., 2021), which aligns with our findings.

Note that the average contributions over the study period shown in Figure 2 obscured the diel variation patterns of contributions on individual days. We have examined the day-by-day variations of cooking emission tracers and oxygenated cooking OA and found that they have peak concentrations during both lunchtime and dinnertime on many days. Figure R1 shows five representative days in which the concentrations of cooking emission tracers (unsaturated fatty acids) and oxygenated cooking OA (azelaic acid) and their apportioned-ON peaked during

both lunchtime and dinnertime. Therefore, the expected enhanced cooking emission and oxidation during lunchtime have been observed; however, on an average basis, more remarkable contributions of cooking emission and oxidation to ON were found during dinnertime compared to lunchtime. Figure R1 has been added as Figure S8 in the revised manuscript.

We have added more discussions in the text as following:

> Line 278-287: "It is noted that the study-wide average contributions to ON from cooking emission and their oxidation product factors were much lower at noontime compared to evening (Figure 2). This difference might be attributed to the lower cooking emissions and/or more favorable dispersion conditions during lunchtime. It is also worth noting that the study-wide average contributions shown in Figure 2 obscure the diel variation patterns on individual days. To address this, we examined the day-by-day variations of cooking emission-related tracers. On many days, both unsaturated fatty acids (indicative of primary cooking emissions) and azelaic acid (a marker of oxygenated cooking OA) exhibited bimodal peaks during lunchtime and dinnertime. Figure S8 highlights five representative days in which the concentrations of these tracers, along with their associated-ON contributions, clearly showed maxima at both mealtime periods."

[Figure]

Figure R1 (the same as newly added Figure S8). Five representative cases in which the concentrations of cooking emission tracers (unsaturated fatty acids) and oxygenated cooking OA (azelaic acid) and their apportioned-ON peaked during both lunchtime and dinnertime.

*3) For Lines 232-233. Based on my understanding, sulfates, isoprene, and α-pinene can all contribute to the formation of ON (e.g., sulfate significantly promotes the formation of organic amine salts; the sulfates and isoprene-related nitrooxy-OS formation exist…), but the authors*

*suggested that they do not contribute to ON. Does this conclusion require more supporting evidence?*

Response: Thank you for your comment. As the reviewer mentioned, sulfate may promote the formation of ON aerosols through two potential pathways: 1) Neutralization reactions between sulfate and nitrogen-containing organic bases such as amines; 2) Sulfate-involved formation of ON species, such as nitrooxy-OS.

Regarding the neutralization reaction, we note that ammonia/ammonium is abundant in the urban Shanghai atmosphere. As shown in Figure R2, particulate sulfate and nitrate have been fully neutralized by ammonium. Additionally, the molar ratio of ammonium to sulfate was 1.8 for the PMF-resolved sulfate formation processes factor. These results suggest that sulfate and/or nitrate have limited potential to enhance the gas-to-particle (G-P) partitioning of organic bases like amines. Furthermore, the abundant presence of ammonium may inhibit the formation of ON species via acid-catalyzed reactions.

In most cases, sulfate concentrations were low (<4 μg m⁻³) and exhibited slight variations throughout the observation period. However, ON concentrations varied significantly over the period (Figure 1). This suggests that sulfate was not a major driver of ON variations. This argument is further supported by the poor correlation ($R^2$ = 0.05) between sulfate and ON concentrations in this study.

The potential contribution to ON aerosol from organic nitrate formation involving the reactions between BVOCs (oxidation products) and $NO_x$/$NO_3$ radical might have been captured within the nitrate formation processes factor. In this study, we found a substantial fraction (14%) of ON was distributed in the factor of nitrate formation processes. As we proposed in the manuscript: "The ON associated with nitrate formation processes may indicate the formation of organic nitrates, which share common precursor of $NO_x$ with nitrate." ON has stronger correlations with $NO_x$ ($R^2$=0.45) and nitrate ($R^2$=0.39) compared to BSOA tracers ($R^2$=0.18), which might be the reason that organic nitrates were apportioned to the factor of nitrate formation processes rather than biogenic SOA factors. This result might be linked to the differences in the underlying formation mechanisms of organic nitrates and biogenic SOA tracers, but currently they are not well understood.

We have added further discussion in the main text:

> Line 288-294: "Sulfate formation processes exhibited negligible contributions to ON formation in this study. Note that sulfate and nitrate were fully neutralized by ammonium (Figure S9), limiting their capability to absorb organic bases such as amines and suppress acid-catalyzed ON formation pathways initiated by sulfuric acid. Furthermore, sulfate and ON displayed distinct temporal variation patterns and were poorly correlated ($R^2$ = 0.05), further indicating a lack of mechanistic linkage between sulfate chemistry and ON formation in the urban atmosphere of Shanghai. These findings collectively suggest that sulfate-driven processes played a minor role in ON formation under the conditions observed."

Line 295-304: "SOA formation from isoprene and α-pinene oxidation contributed insignificantly to the observed ON. While it is known that reactions between biogenic volatile organic compounds such as isoprene and α-pinene with $NO_x$ or $NO_3$ radicals can yield organic nitrates, such contributions may have been captured within the nitrate formation factor, rather than the biogenic SOA factors, in the PMF analysis. This interpretation is supported by correlation analysis, where ON showed stronger associations with $NO_x$ ($R^2 = 0.45$) and nitrate ($R^2 = 0.39$) than with isoprene- and α-pinene-derived SOA tracers ($R^2 = 0.18$–$0.19$). The apparent allocation of biogenically derived organic nitrates to the nitrate formation factor may reflect differences in the formation mechanisms of organic nitrates versus those of traditional biogenic SOA tracers. However, the specific chemical pathways governing these processes remain insufficiently understood and warrant further investigation."

[Figure]

Figure R2. Charge balance between ammonium, sulfate and nitrate during the observation. This figure has been added as Figure S9 in the revised supporting information file.

*4) For Lines 234-236. Generally, winter pollution should be heavier, as indicated by many previous studies in Shanghai, especially the impact of coal combustion. However, the authors mentioned that the winter case was consistent with the summer case. This doesn't seem to be in line with the actual situation.*

Response: Sorry for causing the confusion. Here, we aim to argue that vehicle emissions and coal combustion were major primary sources of ON in both summer and fall-winter time. The absolute contributions in fall-winter were indeed higher than in summer. Specifically, the mass contributions of vehicle emissions and coal combustion to ON were 1.7 and 1.5 times higher than those in summer, respectively.

In the revised manuscript, we have rephrased the sentence:

> Line 245-246: "This finding was consistent with the summertime measurements at the same site that vehicle emissions and coal combustion dominated the primary emission sources of ON (Yu et al., 2023)."

*5) For Lines 237-240: Why is there no consistency between the contribution of coal combustion to ON and the contribution of industrial production to ON. This is somewhat confusing.*

Response: Industrial production mainly refers to manufacturing such as motor industry and electronics product manufacturing, while coal combustion is primarily associated with power plants to generate electricity. Industrial emissions and coal combustion represent different emission sources. The former is characterized by high loading of manganese (Mn), iron (Fe), and zinc (Zn), while the latter has characteristic emissions of selenium (Se) and lead (Pb). Note that the contribution of industrial emissions to ON was higher during daytime, which might be due to the more intense manufacturing activities in daytime. While the source strength for coal combustion is expected to exhibit smaller diel variation as the power plant requires continuous operation., we found an elevated contribution of coal combustion to ON at nighttime. This might be due to the lower planet boundary layer height during the nighttime that leads to an elevation of coal combustion pollutants. To summarize, the contribution of industrial emissions to ON is not expected to be consistent with that of coal combustion.

*Overall, if possible, it is recommended that the author consider the validity of the PMF results more.*

Response: We appreciate the reviewer's suggestion and agree that validity of the PMF results must be carefully examined and supported. To ensure the robustness of the analysis, we have run PMF model many times and selected the solution with the most physically interpretable and consistent factor profiles. All the PMF-resolved factors have passed stability tests, confirming the reliability of the source factors. In the revised manuscript, we have also added the correlation matrix between each resolved factor (Table S2), showing their statistical independence. Importantly, as we have presented above, the identified factors are consistent with established understanding of ON sources, and the results offer new insights into the emission sources and formation processes of ON in the urban atmosphere.

*More comments:*

*1.    Lines 35-36 and Lines 235-236: The data seems to be inconsistent, as mentioned in the abstract, the contribution of both coal combustion and vehicle emissions to ON is 21%. The value reported in the main text is 20%.*

Response: Sorry for making the confusion. In the main text, we stated that "Coal combustion and vehicle emissions were the two dominant primary sources of ON, each contributing **approximately** 20% to the aerosol ON". To keep consistency, we have revised "20%" to "21%" in the main text. (Line 244)

*2.     As reported by the author, the contribution of the primary sources to ON in Shanghai during winter was greater than that of the secondary processes. This seems to be a more*

*important conclusion. However, a significant amount of primary ON emissions are also partitioned to the particulate phase through secondary processes. For example, for organic amines from primary sources, the formation of particulate organic amine salts (they are abundant in fine particulate matter) is closely related to sulfates, nitrates, and organic acids. Therefore, I am still very confused as to why sulfate formation did not contribute to the formation of ON in this study.*

Response: First, we would like to clarify that both primary emissions and secondary formation were identified as major sources of ON aerosols in urban Shanghai. The secondary formation of ON aerosols involved several pathways, as discussed in the manuscript. We agree that the gas-particle (G-P) partitioning of amines could contribute to secondary ON, particularly through reactions forming aminium salts with acidic species. As noted in Section 3.3 and supported by Figure S9, sulfate and nitrate were nearly fully neutralized by excess ammonia/ammonium in the Shanghai winter atmosphere. This neutralization likely limited their ability to drive the partitioning of amines into the particle phase via acid-base reactions.

Although organic amines have been highlighted for their roles in new particle formation and growth, previous studies have shown that nitrogen from small amines contributes less than 2% to bulk aerosol ON in most continental regions (Ho et al., 2015; Liu et al., 2017; Yu et al., 2024). In our study, a notable fraction (8%) of ON was associated with dicarboxylic acid (DCA) formation processes. As shown in Section 3.3, DCA-related ON exhibited a positive correlation with ammonium, suggesting that reactions between DCAs and $NH_3/NH_4^+$ may play a role in ON formation. Given the similar emission sources of $NH_3$ and amines, it is also plausible that reactions between DCAs and amines (i.e., organic acid + organic base) contributed to the observed ON, although this mechanism could not be directly ascertained based on our field measurements alone.

Furthermore, it is possible that some organic aminium salts were apportioned to primary emission factors in the PMF analysis. This may occur when amines are primarily emitted and subsequently undergo G–P partitioning to form aminium salts, resulting in correlations with primary source profiles. We acknowledge that PMF analysis has limitations in fully separating primary and secondary contributions, especially when secondary products retain strong source signatures.

We have added further clarification on this point in the revised section to address the reviewer's concern more explicitly.

Section 3.2:

Line 305-309: "In summary, both primary emissions and secondary formation processes were identified as major contributors to ON in the urban atmosphere of Shanghai. It is also important to note, however, that PMF analysis has inherent limitations in fully separating primary and secondary sources, particularly for compounds such as organic aminiums, which may originate from primary emissions but subsequently undergo gas-particle partitioning."

Future research should employ integrated approaches combining source apportionment, molecular characterization, and thermodynamic modeling to better constrain the sources and formation mechanisms of organic aminium aerosols. In particular, the roles of gas-particle partitioning and secondary reactions involving organic acids and ammonia/amines ($NH_3/NH_4^+$) warrant further investigation to elucidate their contribution to ON formation in urban environments.

References:

Ho, K. F., Ho, S. S. H., Huang, R. J., Liu, S. X., Cao, J. J., Zhang, T., Chuang, H. C., Chan, C. S., Hu, D., Tian, L. W.: Characteristics of water-soluble organic nitrogen in fine particulate matter in the continental area of China, Atmos. Environ., 106, 252-261, 2015.

Huang, D. D., Zhu, S. H., An, J. Y., Wang, Q. Q., Qiao, L. P., Zhou, M., He, X., Ma, Y. G., Sun, Y. L., Huang, C., Yu, J. Z., and Zhang, Q.: Comparative Assessment of Cooking Emission Contributions to Urban Organic Aerosol Using Online Molecular Tracers and Aerosol Mass Spectrometry Measurements, Environ. Sci. Technol., 55, 14526−14535, 2021.

Huang, X. F., Zou, B. B., He, L. Y., Hu, M., Prevot, A. S. H., and Zhang, Y. H.: Exploration of PM2.5 sources on the regional scale in the Pearl River Delta based on ME-2 modeling, Atmos. Chem. Phys., 18, 11563–11580, 2018.

Li, R., Wang, Q. Q., He, X., Zhu, S. H., Zhang, K., Duan, Y. S., Fu, Q. Y., Qiao, L. P., Wang, Y. J., Huang, L., Li, L., and Yu, J. Z.: Source apportionment of $PM_{2.5}$ in Shanghai based on hourly organic molecular markers and other source tracers, Atmos. Chem. Phys., 20, 12047–12061, 2020.

Liu, F. X., Bi, X. H., Zhang, G. H., Peng, L., Lian, X. F., Lu, H. Y., Fu, Y. Z., Wang, X. M., Peng, P. A., Sheng, G. Y.: Concentration, size distribution and dry deposition of amines in atmospheric particles of urban Guangzhou, China, Atmos. Environ., 171, 279-288, 2017.

Wang, Q. Q., Huang, X. H. H., Tam, F. C. V., Zhang, X. X., Liu, K. M., Yeung, C., Feng, Y. M., Cheng, Y. Y., Wong, Y. K., Ng, W. M., Wu, C., Zhang, Q. Y., Zhang, T., Lau, N. T., Yuan, Z. B., Lau, A. K. H., Yu, J. Z.: Source apportionment of fine particulate matter in Macao, China with and without organic tracers: A comparative study using positive matrix factorization, Atmos. Environ., 198, 183-193, 2019.

Yu, X., Li, Q. F., Liao, K. Z., Li, Y. M., Wang, X. M., Zhou, Y., Liang, Y. M., and Yu, J. Z.: New measurements reveal a large contribution of nitrogenous molecules to ambient organic aerosol, npj Clim. Atmos. Sci., 7, 72, https://doi.org/10.1038/s41612-024-00620-6, 2024.

Yu, X., Zhou, M., Li, J. J., Qiao, L. P., Lou, S. R., Han, W. Y., Zhang, Z. J., Huang, C., and Yu, J. Z.: First Online Observation of Aerosol Total Organic Nitrogen at an Urban Site: Insights Into the Emission Sources and Formation Pathways of Nitrogenous Organic Aerosols, J. Geophys. Res. Atmos., 128, e2023JD038921, 2023.

Manuscript ID: egusphere-2024-4103

Title: "Significant secondary formation of nitrogenous organic aerosols in an urban atmosphere revealed by bihourly measurements of bulk organic nitrogen and comprehensive molecular markers"

Author(s): Xu Yu, Min Zhou, Shuhui Zhu, Liping Qiao, Jinjian Li, Yingge Ma, Zijing Zhang, Kezheng Liao, Hongli Wang, Jian Zhen Yu

We thank the reviewer for his/her constructive comments. Each of these comments has been addressed and detailed in our point-by-point response given below. The exact comment text from the reviewer is in black and italic style while our response text is in blue and normal format in this document. The revised texts are marked in blue in the highlighted manuscript. The line numbers mentioned below refer to those in the revised manuscript.

*Comments*

*This manuscript presents an investigation into the sources and formation mechanisms of nitrogenous organic aerosols (ON) in an urban atmosphere, leveraging a new analytical technique for bulk ON quantification and high-resolution source apportionment. The study's strengths lie in its measurement approach, comprehensive source marker analysis, and identification of secondary ON formation pathways, including nitrate formation, photochemical oxidation, and DCA-associated reduced-N species. Several critical gaps in scientific rigor, data interpretation, and presentation weaken the manuscript's impact. Key concerns include incomplete source apportionment due to unclear presentation in methodology, insufficient mechanistic evidence for DCA_ON formation. Addressing these issues would significantly strengthen the manuscript's contribution to atmospheric chemistry and air quality research.*

Response: Thanks for the summary of our work and the overall comments. Please see below for our point-by-point response to your concerns.

**1** *The method used for total nitrogen estimation in this study is not commercially available. Additional details regarding the calibration procedure for total nitrogen quantification using this instrument should be provided. The IN and ON datasets should be cross-validated using independent measurements. For instance, it would be important to demonstrate whether water-soluble inorganic nitrate concentrations correlate strongly with the inorganic nitrogen values quantified in this study.*

Response: Please note that the introductions and validations of our new method for quantifying aerosol IN and ON have been detailed in our previous publications. The first paper was published in 2021 (Yu et al., 2021, Simultaneous determination of aerosol inorganic and organic nitrogen by thermal evolution and chemiluminescence detection, Environ Sci Technol, 55, 11579−11589, doi.org/10.1021/acs.est.1c04876). In this paper, the principle of simultaneous determination of aerosol IN and ON has been elaborated.

The accuracy of IN (including $NO_3^--N$ and $NH_4^+-N$) measurement by the new method has been validated by comparing with ion chromatography (IC) measurements. The results indicated that IN quantities measured by the new method and by IC showed excellent agreement, with both the slope and R-squared value of the linear regression being 0.99, and a negligible intercept. Please refer to Yu et al., 2021 for more details of method characterization and validation. Subsequently, the new analytical approach was applied to online measurements of aerosol IN and ON. Please refer to Li et al. (2022) and Yu et al. (2023). To sum up, we have, in our previous work, demonstrated the robustness of aerosol IN and ON measurement using the new method.

In this study, we have referenced our previous work and briefly introduced the new method as shown in the Methodology section. Nevertheless, we agree with the reviewer that more details on the calibration and cross-validation would be helpful. Accordingly, we have added more descriptions in the revised Methodology section.

> Line 139-144: "4-methyl-imidazole served as the calibration standard for C and N measurements, with systematic calibrations conducted twice monthly. Example calibration curves could be found in Yu et al. (2023) … A comparative analysis of aerosol IN concentrations obtained through the new method and those measured by the Monitor for AeRosols and GAses (MARGA) system was presented in Figure S1."

**2** *The terms "Significant" and "significantly" are overused throughout the manuscript (lines 181, 339, 463, 353), yet no quantitative data or statistical evidence (e.g., percentage) are provided to substantiate these claims. For example, when stating that a result is "significant," the authors should clarify how significant it is (e.g., "a 30% increase with 95% confidence") rather than relying on qualitative assertions. Note that the term "significant" carries strong inferential weight in scientific writing; its use in the main text should be reserved for findings supported by rigorous statistical analysis.*

Response: We have read through the manuscript and replaced instances of "significant" with more appropriate expressions where no statistical significance is implied.

> Line 222: "significant presence" → "high loadings"
>
> Line 260: "significant" → "considerable"
>
> Line 271: "significant" → "remarkable"
>
> Line 351: "significantly" → "largely"
>
> Line 426: "significant" → "potential"
>
> Line 452: "significant" → "notable"
>
> Line 462: "significant" → "large"

**3** *The assertion in line 194 that the primary sources of OC and ON "did not change across seasons" is questionable. For example, biomass burning and coal combustion are known to contribute seasonally variable OC/ON contributions. The authors should provide supplemental data or additional evidence (e.g., time-resolved source apportionment results) to support this claim in this study, particularly given evidence from prior studies demonstrating no seasonal shifts of ON and OC.*

Response: Sorry for causing the confusion. Here, we intend to express that the categories of major primary sources of OC and ON did not change across seasons. As revealed from this and our previous study, vehicle emissions and coal combustion were dominant sources of OC and ON in both summer and fall-winter time in urban Shanghai. However, the absolute contributions of vehicle emissions and coal combustion to OC and ON in fall-winter were much higher than in summer.

Biomass burning might be enhanced during fall-winter time in many regions of China, as revealed by previous studies. In this study, the PMF analysis resolved a primary biomass burning factor and an aged biomass burning factor (i.e., the nitrocatechol formation processes factor). The biomass burning-related ON was apportioned to the aged biomass burning factor. We have discussed the contributions of (aged) biomass burning to ON in the manuscript:

> Line 249-255: "Biomass burning has been recognized as an important source of aerosol ON (Mace et al., 2003; Chen and Chen, 2010; Yu et al., 2017). In this study, we found that a negligible fraction of ON was apportioned to the primary biomass burning factor while a notable presence (7.2%) of ON in the factor of nitrocatechol formation processes (Figure 2). Nitrocatechols are likely formed through atmospheric reactions between $NO_x$ and catechols emitted from biomass burning. This secondary formation pathway was supported by the good correlation between levoglucosan and nitrocatechols (Figure S7)."

In the revised manuscript, we have rephrased the sentence:

> Line 196-197: "Since the categories of major primary sources of OC and ON did not change much at the urban site over seasons, which is discussed below…"

**4** *The uncertainty associated with the 18 identified factors requires further verification. (a) The correlation matrix between factors should be presented to evaluate their interdependence.*

Response: We have created a correlation matrix between the resolved factors. The matrix table is displayed below and added as Table S2 in supporting information. We have also added more discussions in text S1 in the supporting information: "There were no strong correlations ($R \geq 0.7$) between the resolved factors, indicating that overall, the 18 factors were independent of each other and represented 18 distinct sources."

Table R1. Correlation (R) matrix between the PMF-resolved factors.

| | β-caryophyllene SOA formation | Nitrocatechol formation | Phthalic acid formation | Nitrophenol formation | Vehicle emission | Dicarboxylic acid formation | Cooking emission | Residue oil combustion | Photochemical formation | Sea salt emission | Biomass burning | Coal combustion | Industrial emission | Nitrate formation | Sulfate formation | Soil dust | Isoprene&α-pinene SOA formation | Oxygenated cooking OA |
|---|---|---|---|---|---|---|---|---|---|---|---|---|---|---|---|---|---|---|
| β-caryophyllene SOA formation | 1.00 | | | | | | | | | | | | | | | | | |
| Nitrocatechol formation | 0.29 | 1.00 | | | | | | | | | | | | | | | | |
| Phthalic acid formation | 0.24 | 0.30 | 1.00 | | | | | | | | | | | | | | | |
| Nitrophenol formation | 0.07 | 0.65 | 0.21 | 1.00 | | | | | | | | | | | | | | |
| Vehicle emission | 0.43 | 0.58 | 0.12 | 0.47 | 1.00 | | | | | | | | | | | | | |
| Dicarboxylic acid formation | 0.37 | -0.10 | 0.23 | -0.22 | -0.08 | 1.00 | | | | | | | | | | | | |
| Cooking emission | 0.27 | 0.43 | 0.16 | 0.28 | 0.59 | -0.08 | 1.00 | | | | | | | | | | | |
| Residue oil combustion | 0.26 | 0.20 | 0.01 | 0.23 | 0.32 | 0.26 | 0.01 | 1.00 | | | | | | | | | | |
| Photochemical formation | -0.17 | -0.41 | -0.06 | -0.45 | -0.63 | 0.21 | -0.37 | -0.06 | 1.00 | | | | | | | | | |
| Sea salt emission | -0.18 | 0.12 | -0.08 | 0.32 | 0.07 | -0.26 | 0.06 | -0.14 | -0.14 | 1.00 | | | | | | | | |
| Biomass burning | 0.35 | 0.20 | 0.23 | 0.03 | 0.23 | 0.26 | 0.06 | 0.13 | -0.40 | -0.13 | 1.00 | | | | | | | |
| Coal combustion | 0.22 | 0.54 | 0.14 | 0.50 | 0.42 | -0.01 | 0.14 | 0.30 | -0.43 | 0.05 | 0.37 | 1.00 | | | | | | |
| Industrial emission | 0.29 | 0.43 | 0.12 | 0.34 | 0.56 | 0.03 | 0.33 | 0.24 | -0.21 | 0.13 | 0.00 | 0.40 | 1.00 | | | | | |
| Nitrate formation | 0.32 | 0.48 | 0.42 | 0.32 | 0.37 | 0.15 | 0.23 | 0.21 | -0.22 | 0.05 | 0.23 | 0.44 | 0.33 | 1.00 | | | | |
| Sulfate formation | -0.19 | 0.03 | 0.19 | 0.18 | -0.16 | 0.04 | -0.18 | -0.07 | 0.07 | 0.06 | -0.02 | 0.17 | -0.02 | 0.41 | 1.00 | | | |
| Soil dust | 0.21 | 0.26 | -0.08 | -0.02 | 0.39 | -0.05 | 0.28 | 0.03 | -0.16 | -0.07 | 0.10 | 0.42 | 0.18 | 0.18 | -0.10 | 1.00 | | |
| Isoprene&α-pinene SOA formation | 0.46 | 0.10 | 0.54 | 0.00 | 0.06 | 0.69 | 0.06 | 0.25 | 0.00 | -0.09 | 0.34 | 0.08 | 0.05 | 0.31 | 0.07 | -0.16 | 1.00 | |
| Oxygenated cooking OA | 0.24 | 0.14 | 0.32 | 0.06 | 0.23 | 0.07 | 0.29 | 0.06 | -0.25 | -0.14 | 0.23 | -0.04 | 0.01 | 0.21 | -0.25 | 0.21 | 0.24 | 1.00 |

(b) *The observation that no organic nitrogen (ON) was attributed to biomass burning or isoprene/α-pinene SOA factors (lines 232-233) is unexpected and conflicts with extensive evidence. Biomass burning is a well-documented source of nitrogen-containing organic aerosols (OA) in both rural and urban environments. (Li et al., 2023) Similarly, biogenic VOC (BVOC) oxidation under high-NOx or NO3 radical-dominated conditions has been shown to produce ON (Xu et al., 2014). The authors should reconcile these discrepancies by either (1) re-examining their receptor modeling framework, (2) acknowledging limitations in source apportionment resolution.*

Response: We thank the reviewer's comment. We take this opportunity to clarify that in the resolved PMF source factors, two of them are related to biomass burning (BB), a primary BB factor characterized by high loadings of levoglucosan, mannosan, and galactosan, and an aged BB factor indicated by the dominant presence of nitrocatechols. BB-related ON was apportioned to the aged biomass burning factor.

We have revised the discussions on the contribution of biomass burning to ON in the manuscript:

> Line 249-255: "Biomass burning has been recognized as an important source of aerosol ON (Mace et al., 2003; Chen and Chen, 2010; Yu et al., 2017). In this study, we found that a negligible fraction of ON was apportioned to the primary biomass burning factor while a notable presence (7.2%) of ON in the factor of nitrocatechol formation processes (Figure 2). Nitrocatechols are likely formed through atmospheric reactions between $NO_x$ and catechols emitted from biomass burning. This secondary formation pathway was supported by the good correlation between levoglucosan and nitrocatechols (Figure S7)."

The potential contribution to ON aerosol from organic nitrate formation involving the reactions between BVOCs (oxidation products) and $NO_x$/$NO_3$ radicals might be reflected by the ON mass associated with nitrate formation processes. In this study, we found a substantial fraction (14%) of ON was distributed in the factor of nitrate formation processes. As we proposed in the manuscript: "The ON associated with nitrate formation processes may indicate the formation of organic nitrates, which share common precursor of $NO_x$ with nitrate." ON had stronger correlations with $NO_x$ ($R^2$=0.45) and nitrate ($R^2$=0.39) compared to biogenic SOA tracers ($R^2$=0.18), which might be the reason that organic nitrate went to the factor of nitrate formation processes rather than biogenic SOA factors when conducting PMF analysis. This result might be linked to the differences in the underlying formation mechanisms of organic nitrates and biogenic SOA tracers, but currently they are not well understood.

We have added more discussions in the manuscript:

> Line 295-304: "SOA formation from isoprene and α-pinene oxidation contributed insignificantly to the observed ON. While it is known that reactions between biogenic volatile organic compounds such as isoprene and α-pinene with $NO_x$ or $NO_3$ radicals can yield organic nitrates, such contributions may have been captured within the nitrate formation factor, rather than the biogenic SOA factors, in the PMF analysis. This interpretation is supported by correlation analysis, where ON showed stronger associations with $NO_x$ ($R^2 = 0.45$) and nitrate ($R^2 = 0.39$) than with isoprene- and α-pinene-derived SOA tracers ($R^2 = 0.18$–$0.19$). The apparent allocation of

biogenically derived organic nitrates to the nitrate formation factor may reflect differences in the formation mechanisms of organic nitrates versus those of traditional biogenic SOA tracers. However, the specific chemical pathways governing these processes remain insufficiently understood and warrant further investigation."

We acknowledge uncertainties of PMF modeling in apportioning sources to ON aerosol, despite we tried our best to obtain a "most reasonable" PMF solution. We have acknowledged the uncertainties in the "Conclusion and implications" section in the manuscript.

(c) *The proposed formation pathway for ON from oxidized cooking emissions lacks mechanistic clarity and experimental support. The authors should either (1) provide chemical speciation data (e.g., HRMS) linking cooking-derived organic compounds to ON precursors, (2) reference chamber studies demonstrating this pathway, or (3) propose a plausible reaction mechanism (e.g., radical-induced coupling of cooking-derived VOCs with NOx). Without such evidence, this conclusion remains speculative.*

Response: Thanks for the suggestions. Previous chamber studies have suggested that $N_2O_5/NO_3$ radicals could effectively react with unsaturated fatty acid particles from cooking emissions, leading to the formation of organic nitrates (Gross et al., 2009; Zhao et al., 2011). In the revised manuscript, we have added more discussions of ON formation associated with cooking oxidation processes.

Line 275-277: "The formation of ON compounds associated with oxygenated cooking OA likely involves the reactive uptake of $N_2O_5/NO_3$ radicals by unsaturated fatty acid particles, leading to the production of organic nitrates (Gross et al., 2009; Zhao et al., 2011)."

**5** *The strong correlation between DAC_ON and $NH_4^+$ (line 324) requires further scrutiny. While demonstrating agreement between these variables is important, the authors must also rule out potential correlations with other PMF factors (e.g., Nitrate_ON) to confirm the specificity of this relationship.*

Response: Ammonia ($NH_3$), as the most abundant atmospheric base, forms ammonium ($NH_4^+$), which readily associates with acidic species, both inorganic (e.g., nitrate, sulfate) and organic acids (e.g., dicarboxylic acids, DCAs). Therefore, a positive correlation with $NH_4^+$ is not expected to be specific to DCAs alone. Accordingly, our intention was not to use the $NH_4^+$–DCA_ON correlation to suggest a unique or exclusive association, but rather to support the broader hypothesis that DCA-related organic nitrogen (DCA_ON) likely contributes to reduced ON. Additional details addressing this comment are provided below.

We examined the correlations between $NH_4^+$ and other secondary ON factors identified by PMF. $NH_4^+$ showed weak correlation with photochemical formation-ON and oxygenated cooking-ON ($R^2 < 0.1$), a moderate correlation with nitrocatechol formation-ON ($R^2 = 0.19$), and a strong correlation with nitrate-ON ($R^2 = 0.84$). The strong $NH_4^+$– nitrate-ON correlation is primarily attributed to the well-known association between $NH_4^+$ and nitrate ($NO_3^-$). Furthermore, since both nitrocatechols and nitrate originate from $NO_2$,

a moderate correlation ($R^2 = 0.23$) was observed between nitrocatechol and nitrate, which may explain the apparent correlation between nitrocatechol formation-ON and $NH_4^+$. However, a significant correlation does not necessarily imply a direct mechanistic contribution. As discussed in the manuscript, nitrocatechol formation-ON and nitrate-ON are more likely associated with oxidized ON species, such as nitroaromatic compounds and organic nitrates. In contrast, the observed association between DCA_ON and $NH_4^+$ has a more plausible mechanistic basis. Organic acids like DCAs can react with $NH_3/NH_4^+$ to form organic ammonium salts, which may undergo further reactions, such as dehydration, to produce amides—components commonly classified as reduced ON. This mechanistic pathway motivated our analysis of the DCA_ON–$NH_4^+$ relationship. We found a clear positive correlation, which provides observational support for the potential formation of reduced ON species involving DCAs.

While we acknowledge that a positive correlation with $NH_4^+$ is not exclusive to DCA_ON, the combination of observational evidence (detailed in Section 3.3) and mechanistic plausibility strengthens our interpretation that DCA-related ON likely contributes to reduced ON formation.

We have made some modifications to the revised manuscript:

> Lines 361-364: "Overall, while we acknowledge that a positive correlation with $NH_4^+$ is not exclusive to DCA_ON, the combination of observational evidence described above and mechanistic plausibility strengthens our interpretation that the DCA-related ON likely contributes to reduced ON formation."

**6** *The statement in lines 393–396 that higher relative humidity (RH) promotes DCA_ON formation directly contradicts the evidence presented in Figure 3b, which shows that higher pH enhances DCA_ON formation. Higher RH usually leads to higher pH values.*

Response: We apologize for causing this confusion. Here, we have re-examined our results: according to Figure S9 in the supporting information, we found that, overall, the concentrations of DCAs and hDCAs increased with rising RH. However, some extremely high concentrations of DCAs and hDCAs were also observed under moderate RH conditions (50-70%), suggesting the influence of other contributing factors. Figure 3 shows the correlations between DCA_ON and $NH_4^+$ during periods where DCA_ON exhibited a continuous increase over a few hours. These cases were grouped into higher slope cases and lower slope cases. A higher-slope value indicated a more rapid formation of DCA_ON at a given concentration of $NH_4^+$. We found that the averaged aerosol liquid water (ALW) content, estimated using the thermodynamic equilibrium model ISORROPIA II with MARGA data, was much lower in higher slope cases (11.3 μg m$^{-3}$) compared to lower slope cases (49.6 μg m$^{-3}$). This finding is consistent with the lower pH values in higher slope cases.

The occurrences of higher and lower slope cases appeared to be mainly attributed to the differences in atmospheric oxidation capacity, which led to substantial variations in concentration of DCAs–the precursors of DCA_ON. The higher-slope cases corresponded to perods with stronger atmospheric oxidation capacity, as indicated by elevated level of ozone. This enhanced formation of DCAs likely accounted for the more rapid formation of DCA_ON. In comparison, RH and pH did not explicitly facilitate the

rapid formation of DCA_ON.

In the revised manuscript, we have re-organized the elaboration. To avoid confusion, we have removed the discussions of influences of RH and pH on DCAs / DCA_ON formation. Specifically, we deleted the sentences of "In this study, the concentration of DCAs increased with rising relative humidity (RH) (Figure S9), suggesting a higher RH could enhance the aqueous formation of DCAs and/or the gas-to-particle partitioning of DCAs" and "Additionally, the higher-slope cases were associated with lower pH values (higher acidity) as calculated using the thermodynamic equilibrium model ISORROPIA II with MARGA data (Figure 3b). The lower pH facilitated the gas-to-particle partitioning of $NH_3$ and subsequent reactions involving DCAs and $NH_3$."

We have replaced the comparison of pH between the higher and lower slope cases in Figure 3b with a comparison of ΔDCAs (the differences in DCAs concentrations at the beginning and end of each case) for the two groups. Accordingly, we have added the following discussion to the manuscript:

Line 355-356: "This argument was supported by the overall greater increase in DCA concentrations observed in the higher-slope cases (Figure 3b)".

**7** *Methodology sections switch between past and present tenses (e.g., "We have developed... All measurements were carried out..."). Standardize to past tense for consistency.*

Response: We have revised all present tenses to past tense in the Methodology section when applicable.

References:

Gross, S., Iannone, R., Xiao, S. and Bertram, A. K., Reactive uptake studies of $NO_3$ and $N_2O_5$ on alkenoic acid, alkanoate, and polyalcohol substrates to probe nighttime aerosol chemistry, Phys. Chem. Chem. Phys., 11, 7792–7803, 2009.

Huang, D. D., Zhu, S. H., An, J. Y., Wang, Q. Q., Qiao, L. P., Zhou, M., He, X., Ma, Y. G., Sun, Y. L., Huang, C., Yu, J. Z., and Zhang, Q.: Comparative Assessment of Cooking Emission Contributions to Urban Organic Aerosol Using Online Molecular Tracers and Aerosol Mass Spectrometry Measurements, Environ. Sci. Technol., 55, 14526−14535, 2021.

Li, J. J., Yu, X., Li, Q. F., Wang, S., Cheng, Y. Y., and Yu, J. Z.: Online measurement of aerosol inorganic and organic nitrogen based on thermal evolution and chemiluminescent detection, Atmos. Environ., 271, 118905, 2022.

Li, R., Wang, Q. Q., He, X., Zhu, S. H., Zhang, K., Duan, Y. S., Fu, Q. Y., Qiao, L. P., Wang, Y. J., Huang, L., Li, L., and Yu, J. Z.: Source apportionment of $PM_{2.5}$ in Shanghai based on hourly organic molecular markers and other source tracers, Atmos. Chem. Phys., 20, 12047–12061, 2020.

Li, Y., Fu, T.-M., Yu, J. Z., Yu, X., Chen, Q., Miao, R., Zhou, Y., Zhang, A., Ye, J., Yang, X., Tao, S., Liu,H., and Yao, W.: Dissecting the contributions of organic nitrogen aerosols to global atmospheric nitrogen deposition and implications for ecosystems, National Science Review, 10, 10.1093/nsr/nwad244, 2023.

Xu, L., Guo, H., Boyd, C. M., Klein, M., Bougiatioti, A., Cerully, K. M., Hite, J. R., Isaacman-VanWertz, G., Kreisberg, N. M., Knote, C., Olson, K., Koss, A., Goldstein, A. H., Hering, S. V., de Gouw, J., Baumann, K., Lee, S.-H., Nenes, A., Weber, R. J., and Ng, N. L.: Effects of anthropogenic emissions on aerosol formation from isoprene and monoterpenes in the southeastern United States, Proceedings of the National Academy of Sciences, 112, 37-42, 10.1073/pnas.1417609112, 2014.

Yu, X., Li, Q. F., Ge, Y., Li, Y. M., Liao, K. Z., Huang, X. H. H., Li, J. J., and Yu, J. Z.: Simultaneous Determination of Aerosol Inorganic and Organic Nitrogen by Thermal Evolution and Chemiluminescence Detection, Environ. Sci. Technol., 55, 11579−11589, 2021.

Yu, X., Zhou, M., Li, J. J., Qiao, L. P., Lou, S. R., Han, W. Y., Zhang, Z. J., Huang, C., and Yu, J. Z.: First

Online Observation of Aerosol Total Organic Nitrogen at an Urban Site: Insights Into the Emission Sources and Formation Pathways of Nitrogenous Organic Aerosols, J. Geophys. Res. Atmos., 128, e2023JD038921, 2023.

Zhao, Z. J., Husainy, S., Stoudemayer, C. T. and Smith, G. D., Reactive uptake of $NO_3$ radicals by unsaturated fatty acid particles, Phys. Chem. Chem. Phys., 13, 17809–17817, 2011.

---

## Author Response (AR2)

**Point-by-point response to editor's comments**

Manuscript ID: egusphere-2024-4103

Title: "Significant secondary formation of nitrogenous organic aerosols in an urban atmosphere revealed by bihourly measurements of bulk organic nitrogen and comprehensive molecular markers"

Author(s): Xu Yu, Min Zhou, Shuhui Zhu, Liping Qiao, Jinjian Li, Yingge Ma, Zijing Zhang, Kezheng Liao, Hongli Wang, Jian Zhen Yu

We thank the editor for his comments. Each of these comments has been addressed and detailed in our point-by-point response given below. The exact comment text from the editor is in black and italic style while our response text is in blue and normal format in this document. The revised texts are marked in blue in the highlighted manuscript. The line numbers mentioned below refer to those in the revised manuscript.

*Thanks to the authors for carefully addressing the Reviewers' comments. Both reviewers raised questions regarding the interpolation of PMF results. I believe the additional clarifications and explanations have strengthened the findings. I have a few further suggestions regarding the interpolation of the results:*

*1. The authors analyzed the ion balance by considering NH4+, SO42-, and NO3- to support the assertion that SO42- and NO3- are fully neutralized. However, If there is no typo in the figure or text, a slope of 1.05 for n-NH4 vs (n-SO42-+n-NO3-) suggests that they are not fully neutralized. The authors should consider plotting against 2\*n-SO4+n-NO3 instead. Additionally, the molar ratio of ammonium to sulfate was 1.8 instead of 2. Does this discrepancy indicate possible amines uptake or just quantification uncertainty?*

Response: Thanks for the careful check. We apologize for the typo. The X axis of Figure S9 should be $2*n\text{-}SO_4^{2-}+n\text{-}NO_3^-$ instead of $n\text{-}SO_4^{2-}+n\text{-}NO_3^-$. We have revised the figure in the revised supporting file.

In the PMF-resolved factor of sulfate formation processes, the molar ratio of $n\text{-}NH_4^+$ / $n\text{-}SO_4^{2-}$ was 1.8. The measurement results of $NH_4^+$, $SO_4^{2-}$, and $NO_3^-$, however, revealed that $n\text{-}NH_4^+$ was nearly equal to $2*n\text{-}SO_4^{2-}+n\text{-}NO_3^-$, suggesting 1 molar $SO_4^{2-}$ was typically bound with 2 molar $NH_4^+$. The slight discrepancy (1.8 vs 2.0) might be mainly attributed to the uncertainty in the PMF analysis. Overall, our observation results suggested the uptake of amines by sulfate was limited in urban Shanghai due to the abundant presence of ammonia, as stated in the manuscript.

*2. Lines 295-304 in the revised manuscript. If we carefully look at the factor profile of the nitrate formation process, we do see isoprene SOA tracers there. Previous studies have already pointed out that traditional isoprene SOA tracers, such as 2-methyltetrol and 2-methylerythritol are formed from NOx-free oxidation pathway, whereas 2-methylglyceric acid, which often appears*

*at lower concentrations comes from a NOx-involved pathway. It would be worthwhile to determine which of these tracers was assigned to this factor. The absence of direct ON contribution from BVOCs may be due to the compounds used for PMF not including typical organic nitrates produced from BVOCs. There are quite a lot of studies focusing on SOA and ON formation from isoprene/monoterpenes via different oxidation pathways. My feeling is that this study is not specifically designed for exploring such objects. I suggest removing the last sentence. Furthermore, the first sentence contradicts the subsequent explanations. The contribution may be assigned to the nitrate formation process factor, as stated.*

Response: Thank you for your comment. In the PMF analysis, the isoprene SOA tracer was defined as the sum of 2-methylglyceric acid, 2-methylthreitol, 2-methylerythritol, cis-MTB1, MTB2, and trans-MTB3, as detailed in Table S1 of the supporting file. We concur with the editor that the absence of ON in the BVOC SOA factor could stem from the omission of BVOC-derived organic nitrates as PMF inputs. Instead, ON components generated from BVOC oxidation in the presence of $NO_x$ may have been attributed to the nitrate formation processes. Future studies should prioritize the identification and quantification of ON species with source-specific markers to enhance the source apportionment of ON aerosols.

In the revised manuscript, we have removed the last sentence of the paragraph (Lines 295–304) and modified the opening sentence to:
"Insignificant ON was apportioned to the factor representing SOA formation from isoprene and α-pinene oxidation." (Line 295-296)

*3. Line 253, the phrase "through atmospheric reactions between NOx and catechols" sounds wired. Anyhow, Most VOCs cannot directly react with NOx. This should be revised for clarity.*

Response: We have revised the sentence to "Nitrocatechols are likely formed through the nitration of catechols emitted from biomass burning." (Line 253)